# Proven Fatal Invasive Aspergillosis in a Patient with COVID-19 and *Staphylococcus aureus* Pneumonia

**DOI:** 10.3390/jof7030230

**Published:** 2021-03-19

**Authors:** Frank van Someren Gréve, Romy du Long, Raju Talwar, Charlotte J. P. Beurskens, Huibertus J. Voerman, Karin van Dijk

**Affiliations:** 1Department of Medical Microbiology, Amsterdam University Medical Centers, 1105 AZ Amsterdam, The Netherlands; K.vanDijk1@vumc.nl; 2Department of Pathology, Amsterdam University Medical Centers, 1105 AZ Amsterdam, The Netherlands; r.dulong@amsterdamumc.nl; 3Department of Internal Medicine, Amstelland Hospital, 1186 AM Amstelveen, The Netherlands; r.talwar@zha.nl (R.T.); b.voerman@zha.nl (H.J.V.); 4Department of Anesthesiology, Amsterdam University Medical Centers, 1105 AZ Amsterdam, The Netherlands; c.j.beurskens@amsterdamumc.nl; 5Department of Intensive Care, Amsterdam University Medical Centers, 1105 AZ Amsterdam, The Netherlands; 6Department of Intensive Care, Amstelland Hospital, 1186 AM Amstelveen, The Netherlands

**Keywords:** COVID-19, SARS-CoV-2, Invasive Aspergillosis, *Aspergillus fumigatus*

## Abstract

There is increasing attention for opportunistic pathogens such as *Aspergillus fumigatus* complicating SARS-CoV-2 infections in the critically ill. For invasive fungal disease, establishing a clear diagnosis can be challenging due to the invasiveness of diagnostic procedures required for a proven case. Here we present one of the first proven cases of COVID-19-associated pulmonary aspergillosis by positive culture of post-mortem lung biopsy.

## 1. Introduction

Invasive pulmonary aspergillosis complicating severe viral pneumonia has been established as a clinical entity for influenza virus infections [1], and has been described in severe acute respiratory syndrome coronavirus 1 (SARS-CoV-1) infections in 2003 [2]. Increasingly, there are reports of possible and proven invasive fungal disease secondary to SARS-CoV-2 infection [3,4,5,6]. It is hypothesized that tissue damage and immune-dysregulation due to severe disease predispose patients to opportunistic infections with fungi [7]. As it has been associated with excess mortality [8,9,10], an international group of experts including the European Confederation of Medical Mycology, and International Society for Human Animal Mycology advise to start treatment even in patients with possible suspected fungal infection [11]. Here we report a case of a patient with a severe SARS-CoV-2 infection, which was complicated by *Staphylococcus aureus* pneumonia and proven invasive pulmonary infection with *Aspergillus fumigatus*.

## 2. Case Report

The patient was a 79-year-old man with a history of diabetes mellitus type 2, hypertension, paroxysmal atrial fibrillation and chronic heart failure, who presented at the emergency department of a community hospital in Amstelveen, the Netherlands. He had been tired and short of breath for the last two weeks, with increasing dyspnea and confusion for 1 day. At presentation he was disoriented and hypoxic, had a heart rate of 100/min and mean arterial pressure of 60 mmHg. Chest X-ray showed infiltrates bilaterally. Laboratory results showed a C-reactive protein of 100 mg/L, leucocytes of 10 × 10^9^/L and a lactate acidosis with acute kidney failure. The SARS-CoV-2 PCR on a nasopharyngeal swab tested positive. Due to respiratory insufficiency, he was immediately admitted to the intensive care unit for intubation and invasive ventilation and hemodynamic support with norepinephrine. Intravenous ceftriaxone and ciprofloxacin were started empirically, the latter switched to flucloxacillin when sputum cultures showed growth of 10^4^–10^5^ CFU of *S. aureus*. 

Four days after admission, the condition of the patient deteriorated, requiring increasing dosage of vassopressors and respiratory support, and showed decreased kidney function. Hydrocortisone treatment of 200 mg/day and meropenem were started, and ventilation in prone position and continuous renal function replacement therapy were initiated. In the following days, the clinical condition of the patient improved; prone position ventilation was discontinued after two days, norepinephrine and renal replacement therapy could be discontinued. The hydrocortisone dose was lowered after four days.

However, respiratory support could not be weaned below an inspired oxygen fraction of 45%, and the patient was transferred to a tertiary care hospital 14 days after admission. Here, the patient again deteriorated, requiring increasing respiratory support and continuous renal function replacement therapy had to be restarted. Chest X-ray showed bilateral consolidations, more outspoken in the right lung. Sputum cultures showed growth of *A. fumigatus*, the serum galactomannan index was 1.7, serum 1,3-ß-D-glucan 28 picogram/mL, and anidulafungin and voriconazole were started next to vancomycin and ceftazidime. Subsequent non-bronchoscopic broncho-alveolar lavage (BAL) had a galactomannan index of >7.9, and showed growth of *A. fumigatus*. Molecular resistance typing showed no resistance associated mutations for azoles. Agar-based screening using VIPcheck (Radboud University Medical Center, Nijmegen, the Netherlands [12]) showed no phenotypic resistance to azoles. Despite therapy, the patient showed increasing hypoxia, hypercapnia and hypotension. Further medical treatment was deemed futile by the attending care team, and it was decided to abstain for further treatment. The patient died 17 days after admission. Consent for autopsy was provided; macroscopically, bilateral lung edema was found, with multiple white-yellow consolidations, and multiple lung emboli in the right lung artery. Both lungs showed abscess filled with pus up to a diameter of 6.5 cm (Figure 1A,B). Cultures of two separate biopsies of these lesions showed growth of *A. fumigatus*, but not of *S. aureus*. Histologically, there was a profound congestion, edema and intra-alveolar hemorrhage with formation of hyaline membranes, interstitial inflammation and fibroblastic proliferation throughout the lungs. These findings are consistent with severe diffuse alveolar damage (Figure 2A).

In addition, sampling from abscesses showed collections of neutrophils and necrotic tissue inside the cavities, and hemorrhagic infarction of the surrounding lung parenchyma (Figure 2B). No *Aspergillus* hyphae could be detected in PAS-diastase and Grocott stains.

## 3. Discussion

We present one of the first cases of proven invasive pulmonary aspergillosis in a patient with COVID-19. Several case series show that in some patients with COVID-19, fungi such as *Aspergillus* species, have been cultured from respiratory samples [7,9,13,14]. It remains however challenging to distinguish fungal colonization from invasive infection in immunocompetent patients, even if molds are cultured from BAL fluid. The consensus diagnostic criteria for invasive aspergillosis by the European Organization for the Research and Treatment of Cancer/Mycoses Study Group Education and Research Consortium, are often not met in previously immunocompetent critically ill patients, due to lack of host factors, confounding factors on chest imaging, and the inability to perform lung biopsies in invasive ventilated patients [15]. Recently, a new consensus statement has been proposed on diagnostic criteria of invasive fungal disease in COVID-19 [11] As far as we know, only six proven cases have been reported thus far, by positive microscopy on lung biopsies [3,4,5]. 

In our patient with diabetes mellitus, immune dysfunction may have been induced by a severe viral and bacterial pneumonia, and further suppressed by high dose steroid treatment. Possibly, abscesses caused by *S.aureus* have further pre-disposed the patient to pulmonary invasion of fungi, but post-mortem lung biopsy cultures remained negative for bacteria due to prolonged antibiotic therapy. Although no septate hyphae could be detected by microscopy, two separate post-mortem cultures were positive for *A. fumigatus*, and taken together with the positive galactomannan and culture with the same species in BAL fluid, we considered this a proven case of invasive pulmonary aspergillosis. This case illustrates that in critically ill patients with COVID-19, invasive fungal disease should be taken into consideration. When an infection is suspected, prompt antifungal therapy should be started [16,17].

## Figures and Tables

**Figure 1 jof-07-00230-f001:**
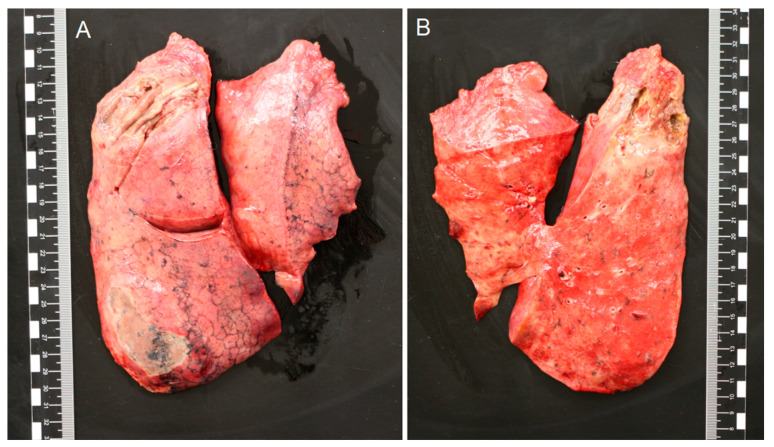
(**A,B**) Macroscopy of the right and left lung. All lobes showed white/yellow consolidations. In the left upper lobe, an abscess with a diameter of 2 cm and in the right upper lobe an abscess of 6.5 cm were found, both filled with green/yellow pus. Cultures of these lesions showed growth of *Aspergillus fumigatus*.

**Figure 2 jof-07-00230-f002:**
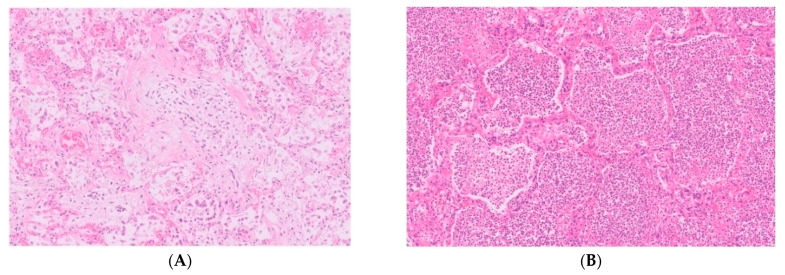
(**A**,**B**). Histological examination (hematoxylin and eosin, ×100) of the lung parenchyma shows profound congestion, edema, interstitial inflammation and fibroblastic proliferation (**A**), and sampling from the abscesses shows large collections of neutrophils inside the alveolar spaces (**B**).

## Data Availability

Not applicable.

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
