# Peer review of "Proven Fatal Invasive Aspergillosis in a Patient with COVID-19 and Staphylococcus aureus Pneumonia"

_jof, 2021, doi:10.3390/jof7030230_

Round 1
Reviewer 1 Report
Dear authors,
thank you very much for this interesting case report. I have the following remarks listed below.
Title
Please write Staphylococcus aureus in italic.
Keywords
Please write Aspergillus fumigatus in italic.
Case report
Line 68. Please write S. aureus in italic.
Line 74-75. There is problem in this sentence, especially line 75 does not fit to line 74. I suppose that you wanted to say that in PAS and Grocott staining no Aspergillus hyphae were detectable.
Were CT scans of the lung performed and if yes, did you see fungal typic infiltrates or cavitations?
I don’t want to search for the fly in the ointment, but for me it is not so clear, if this is a proven case of CAPA or not. If one looks at the publication of Koehler et al. in Lancet Infect Dis. 2020, a proven case is defined by “It is proven by histopathological or direct microscopic detection, or both, of fungal elements that are morphologically consistent with Aspergillus spp, showing invasive growth into tissues with associated tissue damage, or (with or without) aspergillus recovered by culture or detected by microscopy, in histology studies or by PCR from material that was obtained by a sterile aspiration or biopsy from a pulmonary site, showing an infectious disease.”
In your case you could not proof the infection histologically, as you do not find hyphae and associated tissue damage. As you do post-mortem cultivation, the material you culture is not sterile anymore. Pulmonal Aspergillus abscesses in not preformed cavities are not a common disease comparted to pulmonal S. aureusabscesses. Maybe you do not culture S. aureus from the abscesses as it was killed by the multi-antibiotic therapy which was given?! Therefore, this point has to be discussed in much more detail, as it has been done in the manuscript.
Figure 1
Please write Aspergillus fumigatus in italic.
Author Response
Repsonse to Reviewer 1
Title
Please write Staphylococcus aureus in italic.
Thank you. We changed this in the title.
Keywords
Please write Aspergillus fumigatus in italic.
Thank you. We changed this in the Keywords.
Case report
Line 68. Please write S. aureus in italic.
Thank you, we've corrected this (now line 74).
Line 74-75. There is problem in this sentence, especially line 75 does not fit to line 74. I suppose that you wanted to say that in PAS and Grocott staining no Aspergillus hyphae were detectable.
Thank you, we’ve rewritten the sentence to (line 81-82): “No Aspergillus hyphae could be detected in PAS-diastase and Grocott stains.”
Were CT scans of the lung performed and if yes, did you see fungal typic infiltrates or cavitations?
Unfortunately, no CT scan was performed, only chest X-rays. The patient was deteriorating quickly when invasive aspergillosis was considered, and further medical treatment was deemed futile before a CT scan could be made.
I don’t want to search for the fly in the ointment, but for me it is not so clear, if this is a proven case of CAPA or not. If one looks at the publication of Koehler et al. in Lancet Infect Dis. 2020, a proven case is defined by “It is proven by histopathological or direct microscopic detection, or both, of fungal elements that are morphologically consistent with Aspergillus spp, showing invasive growth into tissues with associated tissue damage, or (with or without) aspergillus recovered by culture or detected by microscopy, in histology studies or by PCR from material that was obtained by a sterile aspiration or biopsy from a pulmonary site, showing an infectious disease.”
In your case you could not proof the infection histologically, as you do not find hyphae and associated tissue damage. As you do post-mortem cultivation, the material you culture is not sterile anymore. Pulmonal Aspergillus abscesses in not preformed cavities are not a common disease comparted to pulmonal S. aureus abscesses. Maybe you do not culture S. aureus from the abscesses as it was killed by the multi-antibiotic therapy which was given?! Therefore, this point has to be discussed in much more detail, as it has been done in the manuscript.
Thank you. Indeed, a post-mortem biopsy is not necessarily sterile. However, contamination usually occurs with bacteria, and contamination with Aspergillus would be very unusual. Furthermore, in our case two separate biopsy cultures were positive for A. fumigatus, which was also found in previous BAL fluid cultures. In none of the biopsies bacteria were found. In our view, this makes it very unlikely that the biopsy cultures showing growth of A. fumigatus were positive due to a contamination in this patient. In addition, the EORTC/MSG criteria for proven invasive fungal disease (de Pauw et al, CID 2019) read that aspirations/biopsies should be performed on a ‘normally sterile site’, such as the lungs, and not e.g. sinus cavities. In this patient it is indeed possible that S. aureus has caused pulmonary abscesses, could not be cultured anymore post-mortem due to approximately 2 weeks of antibiotics, and A. fumigatus has secondarily infected the cavities. Taken together, in our view this would classify as a proven case of invasive fungal disease, and hope you agree.
We’ve added the following to the manuscript, line 73-74: “Cultures of two separate biopsies of these lesions (…)”, and line 99-104: “Possibly, abscesses caused by S.aureus have further predisposed the patient to pulmonary invasion of fungi, but post-mortem lung biopsy cultures remained negative for bacteria due to prolonged antibiotic therapy. Although no septate hyphae could be detected by microscopy, two separate post-mortem cultures were positive for A. fumigatus, and taken together with the positive galactomannan and culture with the same species in BAL fluid, we considered this a proven case of invasive pulmonary aspergillosis”
Figure 1
Please write Aspergillus fumigatus in italic.
Thank you, we’ve corrected this in Figure 1.
Reviewer 2 Report
In this manuscript "Proven Fatal Invasive Aspergillosis in a Patient With COVID-19 and Staphylococcus aureus Pneumonia” Frank van Someren Gréve et.al., based on a case report, concluded that in critically ill patients with COVID-19, the invasive fungal disease should be taken into consideration. When an infection is suspected, prompt anti-fungal therapy should be started.
The comments and suggestions for this manuscript are as follows-
The authors should provide a more comprehensive introduction focusing on “Invasive pulmonary aspergillosis complication” with some recently published data and appropriate references.
Author Response
The authors should provide a more comprehensive introduction focusing on “Invasive pulmonary aspergillosis complication” with some recently published data and appropriate references.
Thank you, we’ve rewritten the introduction as follows:
“Invasive pulmonary aspergillosis complicating severe viral pneumonia has been established as a clinical entity for influenza virus infections (1), and has been described in severe acute respiratory syndrome coronavirus 1 (SARS-CoV-1) infections in 2003 (2). Increasingly, there are reports of possible and proven invasive fungal disease secondary to SARS-CoV-2 infection (3–6). It is hypothesized that tissue damage and immune-dysregulation due to severe disease predispose patients to opportunistic infections with fungi (7). As it has been associated with excess mortality (8–10), an international group of experts including the European Confederation of Medical Mycology, and International Society for Human Animal Mycology advise to start treatment even in patients with possible suspected fungal infection (11). (…)”
Reviewer 3 Report
Dear Authors,
The manuscript ID: jof-1137026 entitled „Proven Fatal Invasive Aspergillosis in a Patient With COVID-2 and Staphylococcus aureus Pneumonia” written by Frank van Someren Gréve, Romy du Long, Raju Talwar, Charlotte J.P. Beurskens, Huibertus J. Voerman and Karin van Dijk is devoted to very original case report.
This work is very timely due to the SARS-CoV-2 pandemic. The invasive pulmonary aspergillosis is a severe fungal infection with a high mortality rate. Its incidence is on the rise due to an increase in the number of patients with diabetes mellitus and immune dysfunction. Moreover, extensive studies performed by the Authors proved invasive aspergillosis in the patient with viral and bacterial pneumonia. This case showed that in critically ill patients with COVID, invasive fungal disease and possible prophylactic antifungal therapy should be considered.
It is a well-written and appropriately organized manuscript. I think that it is valuable and worth publishing in “Journal of Fungi”.
With highest regards,
Author Response
We would like to thank the reviewer for stating that our manuscript is valuable and worth publishing in this journal.
Reviewer 4 Report
This case report is interesting from the perspective of the actual pandemic. The only missing aspect I see here is the inclusion of a couple of panels showing the histological features described in this case report.
Author Response
This case report is interesting from the perspective of the actual pandemic. The only missing aspect I see here is the inclusion of a couple of panels showing the histological features described in this case report.
Thank you, we’ve added two photo's of the histological examination (Figure 2A and B).
Round 2
Reviewer 1 Report
Dear authors,
Thank you for the revisions of the manuscript. I agree, even if not fitting 100% of the definition given in Koehler et al, as some findings are from an autopsy, one could conclude that this is proven case based on the following findings: two separate (post mortem) biopsy cultures positive for A. fumigatus, positive galactomannan and A. fumigatus cultured from BAL fluid.
Best regards